

# Inferring [222]Radon soil fluxes from ambient [222]Radon activity and eddy covariance measurements of $CO_2$

S. van der Laan[1], S.N. Manohar[2], A.T. Vermeulen[3,*], F.C. Bosveld[4], H.A.J. Meijer[2], A.C. Manning[1], M.K. van der Molen[5], and I.T. van der Laan-Luijkx[5]

[1]Centre for Ocean and Atmospheric Sciences, School of Environmental Sciences, University of East Anglia, United Kingdom
[2]Centre for Isotope Research, University of Groningen, Groningen, the Netherlands
[3]Energy Research Centre of the Netherlands, Petten, the Netherlands
[4]Royal Netherlands Meteorological Institute, De Bilt, the Netherlands
[5]Meteorology and Air Quality, Wageningen University, Wageningen, the Netherlands
[*]Now at: Dept. Phys. Geography & Ecosystem Science, Lund, Sweden

*Correspondence to:* Sander van der Laan (S.Van-Der-Laan@uea.ac.uk)

**Abstract.** We present a new methodology, which we call Single Pair of Observations Technique with Eddy Covariance (SPOT-EC), to estimate regional scale surface fluxes of [222]Radon ([222]Rn) from tower-based observations of [222]Rn activity, $CO_2$ mole fractions and direct $CO_2$ flux measurements from eddy covariance. For specific events, the regional ([222]Rn) surface flux is calculated from short term changes in ambient ([222]Rn) activity scaled with the ratio of the mean $CO_2$ surface flux for the

5 specific event versus the change in its observed mole fraction. The resulting [222]Rn surface emissions are integrated in time (between the moment of observation and the last prior background levels) and space (i.e. over the footprint of the observations). The measurement uncertainty obtained is about ± 15% for diurnal events and about ± 10% for longer term (e.g. seasonal or annual) means. The method does not provide continuous observations, but reliable daily averages can be obtained. We applied our method to in-situ observations from two sites in the Netherlands: Cabauw station (CBW) and Lutjewad station (LUT).

For LUT, which is an intensive agricultural site, we estimated a mean [222]Rn surface flux of (0.29 ± 0.02) atoms cm[-2] s[-1] with values > 0.5 atoms cm[-2] s[-1] to the south and southeast. For CBW we estimated a mean [222]Rn surface flux of (0.63 ± 0.04) atoms cm[-2] s[-1]. Highest values were observed to the southwest, where the soil type is mainly peat or river-clay respectively. For both stations a good agreement was found between our results and those from measurements with accumulation chambers and two recently published [222]Rn soil flux maps for Europe. At both sites, large spatial and temporal variability of [222]Rn surface fluxes

were observed which would be impractical to measure with an accumulation chamber. SPOT-EC therefore offers an important new tool for estimating region scale [222]Rn surface fluxes and for gaining new insights in the driving mechanisms behind [222]Rn surface emissions. Practical applications furthermore include calibration of process-based [222]Rn soil flux models, validation of atmospheric transport models and performing regional scale inversions of e.g. greenhouse gases via the SPOT [222]Rn-tracer method.



# 1 Introduction

[222]Radon ([222]Rn) is a radioactive noble gas (half-life 3.82 days) that is produced at a constant rate from [226]Radium (half-life 1600 years), which is relatively uniformly distributed in all soils. When released into the atmosphere, [222]Rn is transported and mixed in the atmosphere similar to all other gases emitted from, or close to, the surface. These features make [222]Rn an important

tracer in atmospheric sciences. It has been used as a tracer to study transport processes in the atmosphere (e.g., Liu et al., 1984; Chevillard et al., 2002) and to evaluate or compare the transport component in atmospheric transport models (Dentener et al., 1999; Gupta et al., 2004; Zahorowski et al., 2004). Another highly useful application of [222]Rn is the direct inversion method commonly refered to as the [222]Rn tracer method (Levin, 1987; Schmidt et al., 1996; van der Laan et al., 2014). With this method, the ratio of the [222]Rn surface flux to a measured [222]Rn activity difference over time at a certain observation height

can be applied to calculate the surface flux of another constituent (e.g., $CO_2$) from its concurrently observed mole fraction difference at the same measurement height. In all of these example applications, however, it is essential that the [222]Rn surface flux is well-known. This is especially true for the [222]Rn tracer method as the resulting surface emissions of, for example, $CO_2$ are directly proportional to the assumed regional [222]Rn surface flux. But unfortunately [222]Rn surface fluxes are still poorly known, especially on local and regional scales. One complicating factor is that, although the production of [222]Rn is directly

related to the uniformly distributed Radium content in the soil and therefore relatively well-known, its surface flux is highly sensitive to soil porosity, temperature and soil moisture content. Therefore, the [222]Rn surface flux can be very heterogeneously spread on regional scales (e.g. because of different water table heights) and vary by orders of magnitude within hours because of, for example, rain fall. Recently, several approaches have been applied to quantify [222]Rn surface fluxes: 1) using gamma dose radiation as a proxy for [222]Rn (Szegvary et al., 2007; Manohar et al., in prep.) and 2) modelling the production and

transport of [222]Rn in soils (Hirao et al., 2010; Karstens et al., 2015a). These efforts have provided new tools for studying the driving mechanisms behind the [222]Rn soil flux on relatively large spatial scales. Unfortunately, these methods are limited by the performance of the models, specifically related to the parameterization of the underlying processes, and hence need to be validated independently. Currently, the only two methods to estimate the [222]Rn surface flux directly, is from observations of increasing activities in the soil (Dörr and Münnich, 1987), or in an accumulation chamber which is more representative

for the actual surface flux (Lehmann et al., 2004; Manohar et al., in prep.). The chamber method, however, does not allow for continuous observations because it takes time to flush the chamber, for the concentrations to build-up inside the box and to perform the actual analysis. Furthermore, the method is obviously limited in terms of spatial representation since it only observes the very small soil surface area of the chamber.

In this paper, we propose a novel approach that utilises and combines in-situ measurements of atmospheric [222]Rn activity

and $CO_2$ mole fractions as well as direct $CO_2$ flux from eddy covariance (EC) observations to determine the average [222]Rn surface flux for a relatively large area defined by the footprint of the observations. We applied our method, which we call Single Pair of Observations Technique with Eddy Covariance (SPOT-EC), to data from two measurement stations in the Netherlands and compared our results to two recently published [222]Rn soil flux maps for Europe, as well as to in-situ measurements from accumulation chambers at both sites. In the next section, we explain our method, together with a description of our data sets



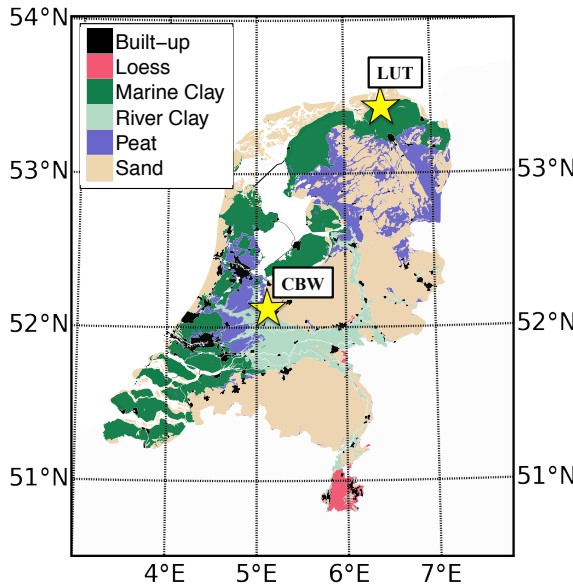

**Figure 1.** Aggregated soil map of the Netherlands developed from the initial soil map by Steur et al. (1985). Also shown are the locations of stations Lutjewad ($53.405°$N, $6.354°$E) and Cabauw ($51.971°$N, $4.927°$E).

used and data selection applied. Our results are described in Sect. 3 followed by a discussion in Sect. 4 and our conclusions in Sect. 5.

## 2    Method

### 2.1    Theory

5    Our methodology for calculating the $^{222}$Rn soil flux is an adaptation of the $^{222}$Rn tracer method (Levin, 1987; Schmidt et al., 1996; Biraud et al., 2000; van der Laan et al., 2009a) where an assumed $^{222}$Rn soil flux is used together with combined observations of ambient $^{222}$Rn activity and, for example, $CO_2$ concentrations at a certain measurement height, to calculate a regional $CO_2$ surface flux. More specifically, we modified the so-called Single Pair of Observations Technique (SPOT) described by van der Laan et al. (2014). This version of the $^{222}$Rn tracer method is more suitable for non-constant surface
10    fluxes and allows for (roughly) selecting the maximum fetch range. The method is based on the concept that all species which are released from, or close to, the surface are transported and diluted in the atmosphere similarly. For example, when the atmosphere is well-mixed, ambient concentrations are observed at (local) background levels and when the atmospheric stability subsequently increases, surface fluxes accumulate within the planetary boundary layer (PBL) and the concentrations increase



as well. The relation between a surface flux and ambient concentrations during such an event of increasing concentrations can be mathematically described as follows (Biraud et al., 2000):

$$\frac{\Delta C_x}{\Delta t} = \int_{t_0}^{t_n} h(t)^{-1} \Phi_x(t) dt = \overline{h^{-1}} \cdot \overline{\Phi_x} \tag{1}$$

here the concentration change of an observed species x over time t ($t_0$ to $t_n$) is given by: $\Delta C_x / \Delta t$ which is the result of its
surface flux $\Phi_x$ accumulating within the PBL and diluted as a function of the mixing height h. Note that both $\Phi_x$ and h(t) are time dependent. The overbar indicates averaging in space (i.e. the footprint) and time, that is, representing the average mixing height and the mean net surface flux during the observation period and for the observed area. Applying Eq. 1 to both $^{222}$Rn and one other gas species, for example, $CO_2$, then taking the ratio of $\Phi^{222}$Rn / $\Phi CO_2$ and rearranging for $\Phi$Rn yields an equation where the mixing height has been cancelled out, namely:

$$\overline{\Phi Rn} = \frac{Rn(t_n) - Rn(t_0)}{CO_2(t_n) - CO_2(t_0)} \cdot \overline{\Phi CO_2} \tag{2}$$

where the resulting $^{222}$Rn soil flux is calculated from the observed concentration changes between local background levels at t=$t_0$ and (a pair of $^{222}$Rn and $CO_2$) observations at t=$t_n$. Equation (2) is basically the inverse of the Single Pair of Observation Technique (SPOT) method described in van der Laan et al. (2014), where instead of using an assumed $^{222}$Rn soil flux to calculate the surface flux of $CO_2$, we use a measured $CO_2$ surface flux (obtained from EC measurements) to calculate the $^{222}$Rn
soil flux. We will refer to this method as SPOT-EC for the remainder of this paper. The term "event" will be used for periods in time that are suitable for applying the SPOT-EC method and further described in Sect. 2.3.

## 2.2 Measurement locations, instrumentation and data used

We applied our methodology on half hourly averaged ambient measurements of the $^{222}$Rn activity and of $CO_2$ mole fractions as well as $CO_2$ surface flux measurements from eddy covariance (EC), at two sites in the Netherlands: Lutjewad (LUT) and
Cabauw (CBW). Both stations are equipped with basic meteorological observations (air temperature, humidity, atmospheric pressure, wind speed and direction and solar radiation) and, via several air intakes on a sampling tower, ambient air is continuously flushed down to a laboratory for further analyses. Station specific information is given below. Figure 1 shows a map of the Netherlands including the main soil types (Steur et al., 1985) and station locations.

### 2.2.1 Lutjewad station

LUT (53.405°N, 6.354°E, 1 m a.s.l.) is a coastal site in the north of the Netherlands about 30 km to the northwest of the city of Groningen (population ~200,000). To the north of the station, with its 60 m tall tower, a reclamation area and tidal flats merge into the North Sea whereas the south sector consists of agricultural area on sea clay soils, see also Fig. 1. The (intensely managed) water table is generally ~1 m below the surface but near-surface during wet periods. The prevailing wind direction





(> 31% of the time) is between 195° and 255° and wind speeds between 6 and 9 m/s are dominant (~35% of the time) at the top intake height at 60 m above ground (van der Laan et al., 2009a). Ambient $CO_2$ mole fractions were measured from a height of 60 m with a modified Agilent HP 6890N Gas Chromatograph (van der Laan et al., 2009b) together with mole fractions of $CH_4$, $N_2O$, $SF_6$, CO. Typically 6 analyses are performed per hour and the measurement precision is about ± 0.08 ppm for $CO_2$.

An eddy covariance system consisting of a LiCor 7500 open path gas analyser and a Gill Windmaster Pro 3-axis ultrasonic anemometer is installed at a height of 50 m for direct surface flux estimates of $CO_2$ (as well as $H_2O$ and sensible and latent heat fluxes) (Dragomir et al., 2012). For the EC $CO_2$ flux measurements at both LUT and CBW, we assumed a measurement uncertainty of about 10% based on Kruijt et al. (2004).

Ambient $^{222}Rn$ activity is measured at both LUT and CBW using a dual-flow loop two-filter detector developed by the

Australian Nuclear Science and Technology organisation (ANSTO) and described by Whittlestone and Zahorowski (1998). Unwanted aerosols and (radioactive) decay products are removed by a filter in front of the detector and the $^{222}Rn$ decay products are sampled on a second filter at the exit of a 1500 litre delay chamber, where their decays are counted by a photo-multiplier. This system uses a non-energy selective alpha particle counter to detect $^{222}Rn$ particles. In principle it also detects 220Rn (half-life of 55.6 s) however this is prevented by the relatively long residence time (~10 half lifes) of the air sample

from the tower inlet to the detector. The measurement precision is about ±5% of the measured value at both sites (Popa et al., 2011; van der Laan et al., 2010). Ambient observations of $CO_2$ mole fractions, $^{222}Rn$ activity and $CO_2$ surface fluxes for the period of Nov 2007 - April 2010 at LUT are shown in Figures 2a to c respectively.

For validation of our method, we use direct measurements of the $^{222}Rn$ soil flux with a soil chamber (surface area ≈ 0.03 $m^2$) located near the foot of the mast. This chamber system, which is described in detail in Manohar et al. (in prep.), uses a

flow-through accumulator method (Zahorowski and Whittlestone, 1996) where the air is continuously circulated between the accumulation chamber and the detector (Lucas Scintillation Cell model 300A + Pylon AB-5 portable radon monitor, Pylon Electronics, Canada). Because of the relatively low radium activity and high soil moisture content at the sites, and given the relatively high detection limit of the Pylon monitor, the chamber needs to accumulate for 4 hours before each measurement which takes 7.5 hours and is followed by a flushing period of 0.5 hours (Manohar et al., in prep.). In this way, two 4-hourly

integrated observations are obtained per day. The soil chamber $^{222}Rn$ measurement uncertainty is estimated at ~ ±20% of the measured value.

### 2.2.2 Cabauw station

CBW (51.971°N, 4.927°E, 0.7 m b.s.l.) is located within a mainly agricultural area about 25 km southwest of the city of Utrecht (population ~340,000), see also Fig. 1. To the south of the station, with its 213 m tall tower, the soil type is mainly river-clay

and to the north mostly peat or peat on clay (Arnold et al., 2010). Within a distance of about 400 m (and up to ~2 km for the WSW sector) the terrain can be classified as open pasture. Similarly to LUT, the (intensely managed) water table is generally ~1 m below the surface but up to near-surface during wet periods. Ambient $CO_2$ mole fractions are measured with a LiCor-7000 non-dispersive infrared analyser sampled from heights of 20 m (used in this study), 60 m, 120 m, and 200 m (Popa et al., 2011; Vermeulen et al., 2011). The measurement precision is generally < ±0.1 ppm. Direct $CO_2$ fluxes are measured at heights





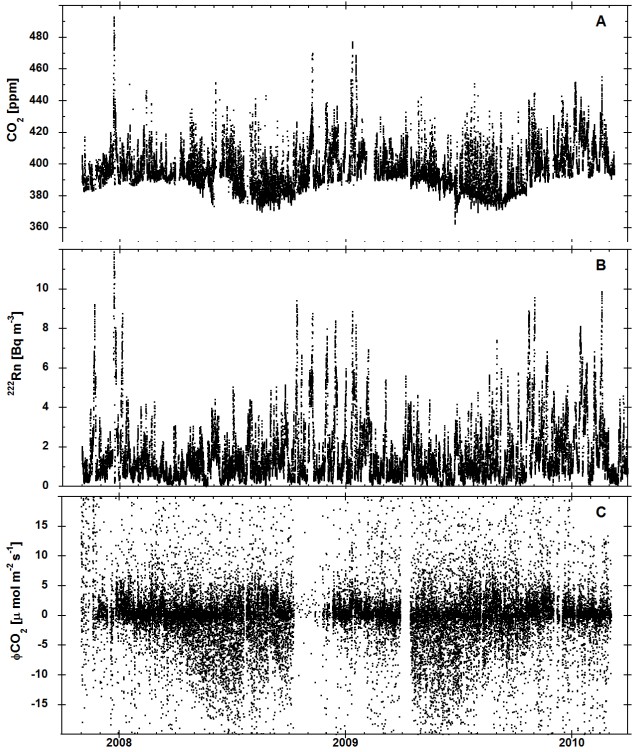

**Figure 2.** Ambient measurements of half hourly averaged $CO_2$ mole fraction (a), $^{222}Rn$ activity (b) and $CO_2$ surface flux (c) from Lutjewad station. X-axis tick marks indicate the beginning of the year stated.

of 3 m, 60 m (used in this study), 100 m and 180 m with a similar EC system as at LUT consisting of a LiCor 7500 open path gas analyser and a Gill R3 ultrasonic anemometer. Because of blockage from the tower, observations during wind directions between 280° and 340° cannot be used. Ambient observations of $CO_2$ mole fraction, $^{222}Rn$ activity and $CO_2$ surface fluxes for the period of Jan 2007 - Jul 2013 at CBW are shown in Figures 3a to c respectively.

## 2.3 Data selection

We selected so-called "events" for both stations according to the (automated) method described by van der Laan et al. (2014). An example for CBW is given in Fig. 4. Events were selected based on the following criteria: the start of an event is detected when at least five out of eight consecutive half-hourly $^{222}Rn$ measurements are higher than the previous measurements, and the first value of the eight (at $t = t_1$) is at least 0.3 Bq m$^{-3}$ higher than the baseline (at $t_0$). Similarly, the end of the event is defined as the time when the maximum value before dropping back to background levels is reached with at least five out of eight consecutive measurements lower than the previous measurement. The $^{222}Rn$ soil flux for the event is calculated with Eq. 2 for each measurement (at $t = t_n$) relative to the local background level at $t = t_0$. EC measurements were processed according to CarboEurope protocols (Aubinet et al., 2000) using EddySoft (Kolle and Rebmann, 2007) for LUT and ALTEDDY software



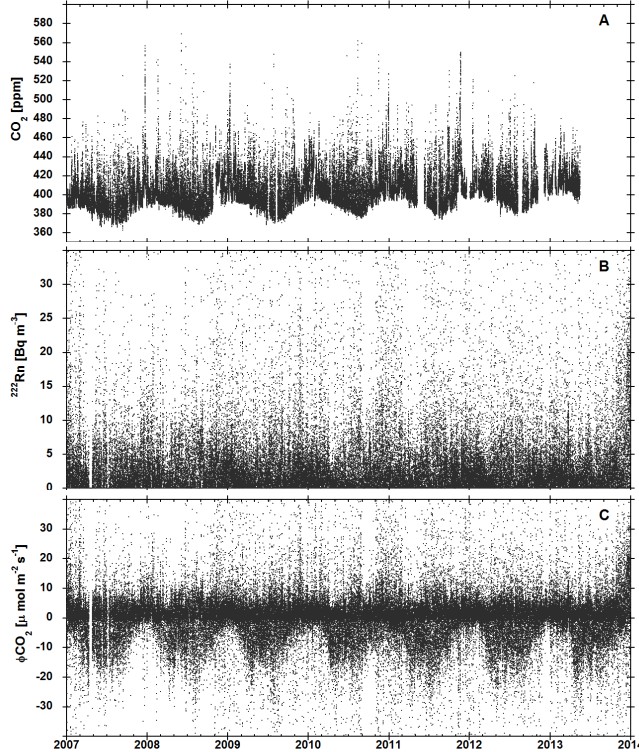

**Figure 3.** Ambient measurements of half hourly averaged $CO_2$ mole fraction (a), $^{222}$Rn activity (b) and $CO_2$ surface flux (c) from Cabauw station. X-axis tick marks indicate the beginning of the year stated.

(www.climatexchange.nl/projects/alteddy/) for CBW. A friction velocity ($u^*$) threshold (Papale et al., 2006) of $> 0.2$ m s$^{-1}$ was determined for both stations and applied to ensure sufficient turbulence for the eddy-dependent EC measurements. Furthermore, the CBW measurements were rejected for wind directions between 280° - 340° because of tower blocking and between 0° - 60° and 240° - 360° in the case of LUT to exclude the marine sector. As a rough strategy to ensure our results are predominantly

5    locally influenced and hence that our EC measurements are represented by the concentration changes of our selected events, results were only accepted for $t_n$ - $t_0$ < 4 hours. Furthermore, a maximum variation in wind direction of 25° was prescribed to ensure stationary conditions during the events. Results were only accepted for dry periods because rain affects the EC measurements of our open path analysers, and finally, results were retained that had a relative uncertainty of $< \pm 75\%$ for CBW and, because of less data, $< \pm 100\%$ for LUT.





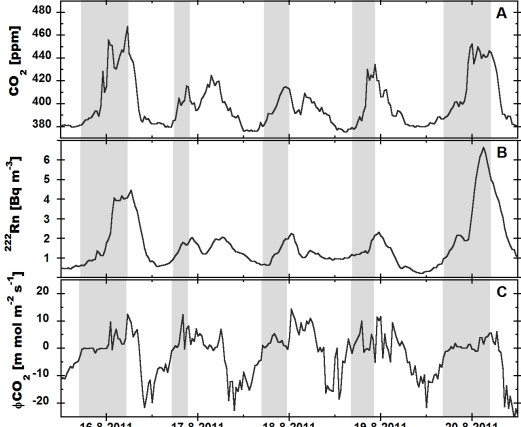

**Figure 4.** An example of diurnal events for CBW of $CO_2$ mole fraction (a), $^{222}$Rn activity (b) and $CO_2$ surface flux (c) for 5 days in August 2011. The events identified by our event selection methodology are indicated by the grey shadings. X-axis tick marks indicate the beginning of the day stated.

## 3 Results

### 3.1 LUT

For LUT, we find a mean $^{222}$Rn surface flux of $(0.43 \pm 0.05)$ atoms cm$^{-2}$ s$^{-1}$ and a median of 0.17 atoms cm$^{-2}$ s$^{-1}$ based on 209 events between Jan 2008 and Jan 2010 (Fig. 5a and Table 1). The error bars on Fig. 5a are calculated from error propagation of

Eq. 2 using the measurement uncertainties described in Section 2.2.1 and 2.2.2. The statistical distribution of the $^{222}$Rn surface fluxes is shown in Fig. 6 (limited to < 2 atoms cm$^{-2}$ s$^{-1}$ for clarity), and from this we find that the mean value is much higher than the median because of a few exceptionally large (i.e. » 1.5 atoms cm$^{-2}$ s$^{-1}$) $^{222}$Rn surface fluxes. After excluding the eleven values > 1.5 atoms cm$^{-2}$ s$^{-1}$, we find a median value of 0.15 atoms cm$^{-2}$ s$^{-1}$ and a mean value of $(0.29 \pm 0.02)$ atoms cm$^{-2}$ s$^{-1}$. The latter is in fact equal to the mean value for the Netherlands of 0.29 atoms cm$^{-2}$ s$^{-1}$ proposed by Szegvary et al. (2007). This

value was taken from a European $^{222}$Rn flux map based on using a gamma dose radiation as a proxy for $^{222}$Rn activity, and has been used in previous studies for this site (van der Laan et al., 2009a, 2010). Note however that the coarse resolution of this map does not allow for any significant distinction between LUT and the mean value for the Netherlands.

Our mean result, even after discounting the eleven high values, is a factor of two higher than the mean value of $(0.16 \pm 0.01)$ atoms cm$^{-2}$ s$^{-1}$ based on soil chamber measurements (Manohar et al., in prep.) and also higher than the model-based estimate

of $(0.19 \pm 0.12)$ atoms cm$^{-2}$ s$^{-1}$ found by Manohar et al. (2013). The measurements from the soil chamber and our SPOT-EC method agree well for the majority of the events, but the higher values are not captured by the chamber method. In these cases, the soil underneath the chamber behaves differently than the average soil in our footprint as seen from the tower. This makes sense as the small chamber only "sees" a single soil type.

**Table 1.** $^{222}$Rn soil flux for CBW and LUT estimated with SPOT-EC, soil chambers and models.

|  | Soil chamber | Model | SPOT-EC (this work) | Units |
|---|---|---|---|---|
| CBW | 0.64 ± 0.09 (mean) | 0.65 ± 0.14[a] | 0.63 ± 0.04 (mean) | atoms cm$^{-2}$ s$^{-1}$ |
|  | 0.62 (median) | 0.59 ± 0.18[b] | 0.34 (median) |  |
|  | N = 14 |  | N = 422 |  |
|  | Period: July 2011[a] |  | Period: Jan 2007 - Jul 2013 |  |
| LUT | 0.16 ± 0.01 (mean) | 0.19 ± 0.12[a] | 0.43 ± 0.05 (mean) | atoms cm$^{-2}$ s$^{-1}$ |
|  | 0.11 (median) | 0.08 - 0.41 ± 0.03[c] | 0.17 (median) |  |
|  | N = 1069 |  | N = 209 |  |
|  | Period: Jun 2008 - Jan 2010[a] |  |  |  |
|  |  |  | 0.29 ± 0.05 (mean)[d] |  |
|  |  |  | 0.17 (median)[d] |  |
|  |  |  | Period: Jan 2008 - Jan 2010 |  |

[a]Manohar et al. (2013).
[b]Values taken from Karstens et al. (2015a) with latitude: 51.54°N, longitude: 4.88°E.
[c]Values taken from Karstens et al. (2015a) with latitude: 53.21°N, longitude: 6.38°E and: latitude: 53.13°N, longitude: 6.38°E.
[d]After excluding 12 values > 1.5 atoms cm$^{-2}$ s$^{-1}$ (see Sect. 3.1).

We also compared our results with results from a recently published process-based $^{222}$Rn flux map for Europe (Karstens et al., 2015a, b). Due to the course resolution of the map (about 50 km x 50 km at the location of our site), our site's location is defined as "sea" in this map and hence $^{222}$Rn fluxes are not available for the exact location of our site. We therefore choose to report the mean values for (1) the first grid cell with $^{222}$Rn fluxes directly to the south (53.21°N, 6.38°E), and (2) the cell

5   below (53.13°N, 6.38°E). For (1) we find a mean value of (0.08 ± 0.03) atoms cm$^{-2}$ s$^{-1}$ and for (2) (0.41 ± 0.03) atoms cm$^{-2}$ s$^{-1}$. The model-based results indicate the values from the grid box closest to the tower are the lowest because of a higher soil moisture content, which is the main driver for the $^{222}$Rn soil flux and a key variable in the model.

This spatial variability of the $^{222}$Rn surface flux is also observed with our measurements and shown in Fig. 7a. This polarplot, generated with the openair package in R, depicts the wind direction versus the maximum fetch range, calculated as wind speed

10   times the duration of the event, versus the $^{222}$Rn surface flux. For clarity, data were limited to > 0.05 atoms cm$^{-2}$ s$^{-1}$ and < 1.5 atoms cm$^{-2}$ s$^{-1}$. In general, the values closest to our tower are around 0.3 atoms cm$^{-2}$ s$^{-1}$. To the southwest and southeast values are observed around >0.5 atoms cm$^{-2}$ s$^{-1}$. The spatial variations are most likely due to different soil or crop types (i.e. affecting the soil moisture content and porosity) since the area around the tower is a very heterogonous agricultural region with rotation of several crop species, open pastures and an intensely managed water table to suit the needs of the agriculture

15   and horticulture. Figure 8a shows the diel distribution of our events versus the magnitude of the calculated $^{222}$Rn surface flux. Although the fraction of the day for which the atmosphere is generally well-mixed (i.e. ~10h - 15h) is under-sampled due too





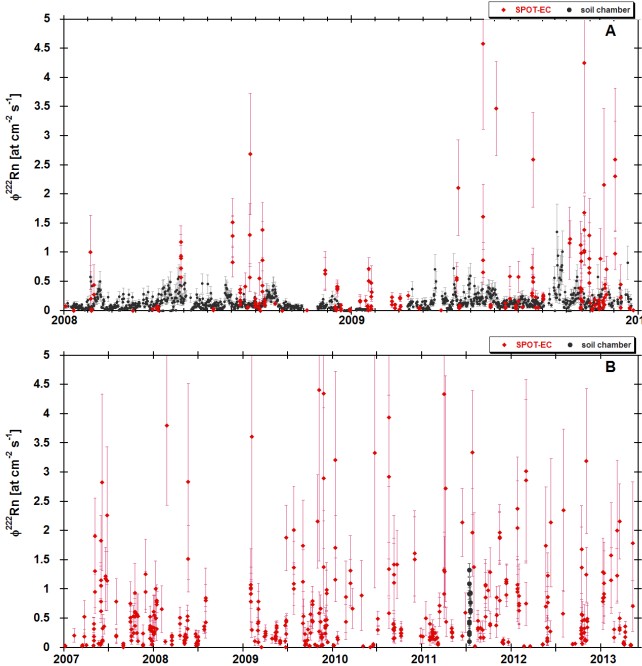

**Figure 5.** $^{222}$Rn surface fluxes calculated with Eq. 2 for LUT (a), and CBW (b). The error bars are calculated from error propagation of the measurement uncertainties as described in Section 2.

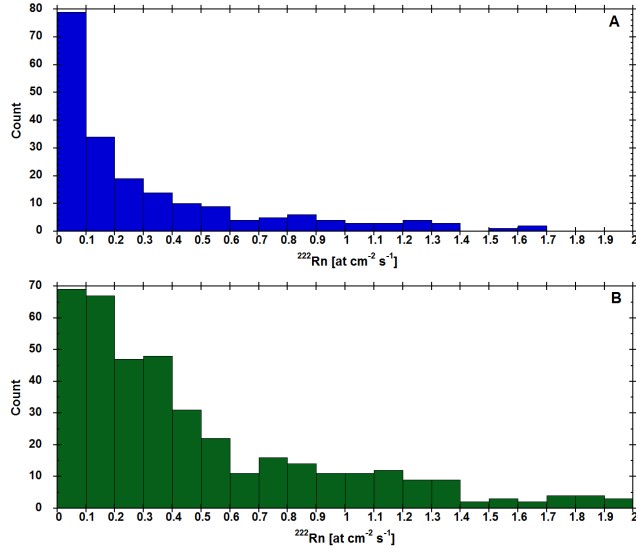

**Figure 6.** Statistical distributions of $^{222}$Rn surface fluxes for LUT (a) and CBW (b). Values > 2 at cm$^{-2}$ s$^{-1}$ (6% of the total) are omitted in the figures for clarity in the case for CBW.



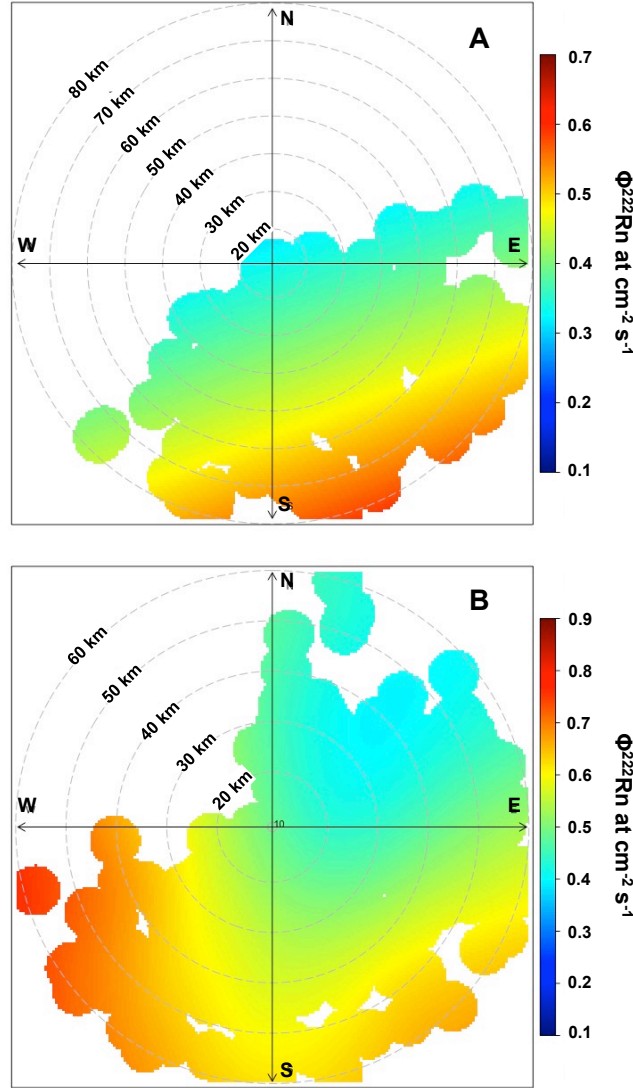

**Figure 7.** Spatial distributions of all analysed $^{222}$Rn surface fluxes for LUT (a) and CBW (b). Plot is generated with the openair package in R. The polar axis indicates the maximum fetch range calculated as wind speed times the duration of the event and is plotted against wind direction and the $^{222}$Rn surface flux. For clarity, data were limited to > 0.05 atoms cm$^{-2}$ s$^{-1}$ and < 1.5 atoms cm$^{-2}$ s$^{-1}$. Note that for LUT, the sector between 240° through 0° to 60° (between WSW and ENE) is not taken into account because of the marine influences, and for CBW the sector between 280° and 340° (WNW-NNW) is blocked by the tower.

small change in concentration, we obtain a reasonable coverage over the day. More importantly, the magnitude of the flux does not seem to be correlated with the time of the day.



## 3.2 CBW

The results for CBW for the period Jan 2007 - Jul 2013 are shown in Fig. 5b and both the mean and median $^{222}$Rn surface flux values are given in Table 1, which also shows values from the soil chamber measurements and model results. The error bars are calculated from error propagation in Eq. 2 using the measurement uncertainties described in Section 2.2.2. The mean value of

$(0.63 \pm 0.04)$ atoms cm$^{-2}$ s$^{-1}$ (n = 422) compares very well with the results from the modelling work of (Manohar et al., 2013) who reported a mean value of $(0.65 \pm 0.14)$ atoms cm$^{-2}$ s$^{-1}$. The results from the process-based model (Karstens et al., 2015a) are in the same range: $(0.59 \pm 0.18)$ atoms cm$^{-2}$ s$^{-1}$.

There is no $^{222}$Rn soil chamber programme at CBW. We organised a short field campaign from 12 - 16 July 2011 (n = 14) with a portable emanometer (Zahorowski and Whittlestone, 1996), the results of which are shown in Fig. 5b. The mean value

of $(0.64 \pm 0.09)$ atoms cm$^{-2}$ s$^{-1}$ compares favourably with the results from our SPOT-EC method, but given the large variability this might of course be accidental. The median value of the SPOT-EC method, 0.34 atoms cm$^{-2}$ s$^{-1}$, is again almost a factor of two lower than the mean, which we attribute to the large variability of the fluxes. The statistical distribution of the regional $^{222}$Rn surface fluxes are shown in Fig. 6b, limited to 2 atoms cm$^{-2}$ s$^{-1}$ for clarity. The fluxes are clearly not normally distributed but rather follow a log-normal shape which is as expected, as the fluxes are uni-directional (van der Laan et al., 2009a). The

large difference between the median and the mean value is a result of the very large temporal variability. Observed $^{222}$Rn surface fluxes can vary by orders of magnitude on hourly to diurnal scales because of changing wind direction or because of rainfall. Figure 7b shows the spatial distribution of our $^{222}$Rn surface fluxes. Although part of this polar plot is masked because of tower blocking, it provides interesting information about the $^{222}$Rn surface fluxes in our footprint. $^{222}$Rn surface fluxes closest to the tower range are on average between 0.4 and 0.6 atoms cm$^{-2}$ s$^{-1}$. Lower values are mostly observed from

the north-east where the soil type is peat or peat on clay. Highest values are mostly observed in the southwest sector where the soil type is mainly river-clay.

The diel distribution of the observed events versus the $^{222}$Rn surface flux is shown in Fig. 8b. Similar to our findings for LUT, well-mixed periods are generally under-sampled but a reasonable coverage over the day is obtained and the magnitude of the flux is not dependent on the time of the day.

## 4   Discussion


The method presented in this paper allows for accurately estimating the $^{222}$Rn surface flux on a regional scale. The flux estimates are integrated in space and time, that is, averaged over the footprint and for the duration of an event (Sect. 2.3). The spatial and temporal range are mostly depending on the sampling height and atmospheric stability but they can also be influenced by the choice of event selection criteria (Sect. 2.3). A coarse estimate based on the length of the selected events

(i.e. $t_n$-$t_0$ in Eq. 2) and the mean wind speed yields a mean fetch range of ∼60 km for LUT and ∼45 km for CBW for our observations. Within these footprints, both sites are relatively heterogeneous with respect to the soil type. Unfortunately our experimental setup was not ideal since, due to practical limitations, at both sites the EC measurements are not taken at the same intake height as the $CO_2$ mole fraction and $^{222}$Rn activity measurements. In some cases therefore, an increase or decrease



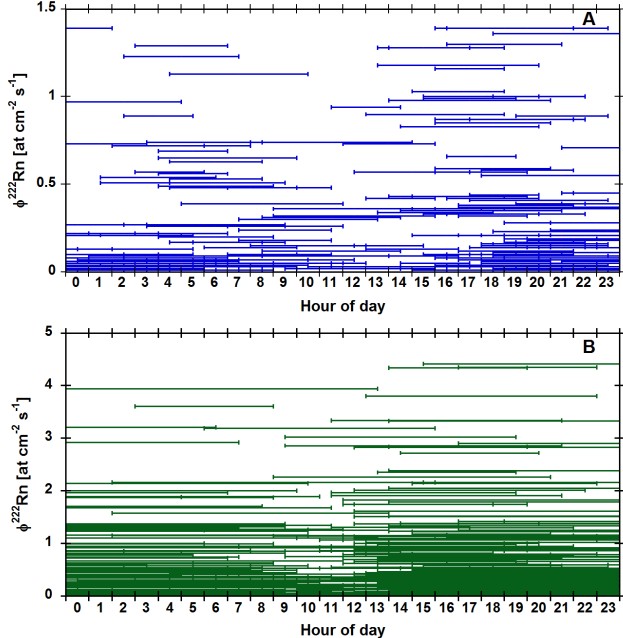

**Figure 8.** Distribution of analysed events over the day for LUT (a) and CBW (b). Length and position of each line indicates the timing and duration of each event. No significant correlation is observed between the sampling time and the magnitude of the flux. Well-mixed periods are generally under-sampled but a reasonable coverage over the day is obtained.

in $CO_2$ mole fraction might not be fully reflected by the EC measurements, especially for a heterogeneous area such as LUT. Similar to the $^{222}$Rn tracer approach as it is commonly applied (e.g. Schmidt et al., 1996; Biraud et al., 2000; van der Laan et al., 2010), we used activity and mole fraction measurements of $^{222}$Rn and $CO_2$, respectively, from the same intake height to ensure any change caused by atmospheric transport and dilution is equally reflected in the measurements of both species. This

is also the main assumption of the $^{222}$Rn tracer method. Since the measurement uncertainties are relatively low, any uncertainty in the calculated flux ($^{222}$Rn in our case) is directly related to the assumed a-priori flux (the EC measured $CO_2$ flux in our case). Typically, where either the a-priori flux is assumed to be well-known or, as in our case, its measurements uncertainty is relatively low, any inaccuracy in the calculated flux is most likely to be an offset and hence can be easily corrected when new information on the a-priori flux becomes available. We have confidence in our $^{222}$Rn flux results calculated with this new

method, because they are in very good agreement with those from three other, independent methods including model-based results and observations from accumulation chambers (Table 1).

   Although we calculate the $^{222}$Rn soil flux from semi-continuous mole fraction, activity and EC measurements, our method does not provide semi-continuous results. This is because the EC systems require relatively turbulent conditions (by definition), whereas the relative uncertainty of the measured concentration changes (i.e. numerator and denominator in Eq. 2) decrease

with increasing concentration changes and hence relative stable conditions. Fortunately, Fig. 8 shows we do have a good



data coverage throughout our observation period. More importantly, there does not appear to be any correlation between the magnitude of the $^{222}$Rn surface fluxes and the time of the day. This suggests that our methodology can be applied to first determine the mean regional $^{222}$Rn surface flux for a site and subsequently to use these fluxes to calculate the regional emissions of other atmospheric gases of interest via the SPOT method. At both sites, the $^{222}$Rn surface flux can vary by orders

of magnitude on hourly to diurnal scales. Most of this variability can be attributed to meteorological conditions (e.g. rainfall or a sudden change in wind direction), or spatial and temporal variability in the soil moisture content and settlement of the soil. The latter affects the porosity and permeability of the soil which control the diffusion of $^{222}$Rn within the soil to the soil-atmosphere interface. This would explain why our chamber and SPOT-EC results at LUT disagree to some extent, in particular that the high $^{222}$Rn surface fluxes of SPOT-EC are not found in the chamber measurements. LUT is an intensive agricultural

site with an artificially controlled water table height. The soil chamber at LUT is placed on undisturbed soil, whereas our regionally integrated results from the SPOT-EC method are influenced by a large, regularly disturbed (ploughed) agricultural area with varying permeability and porosity. Another potential reason for the discrepancy between the very local and regionally integrated $^{222}$Rn surface fluxes is the use of $^{226}$Ra-containing phosphate fertilizer (Feichter and Crutzen, 1990). For example, Dörr (1984) measured a doubling of $^{222}$Rn from intensively used agricultural soils. Contrary to the chamber method, our

SPOT-EC approach captures such variability integrated over a large area. This is a key advantage of our method and allows for follow-up studies targeting the driving mechanisms of $^{222}$Rn surface fluxes.

Even though the $^{222}$Rn surface flux can vary by orders of magnitude on hourly to diurnal scales, the longer term (e.g. seasonal - annual) mean can be estimated with relative high accuracy. In principle, the uncertainty for each individually observed flux can be calculated relatively straightforwardly by error propagation of the measurement uncertainties for each variable. In general,

the fluxes calculated from the largest concentration changes (Eq. 2) have the smallest uncertainty. For LUT, the uncertainties ranged from $\pm17\%$ to $\pm100\%$ with a mean of $\pm42\%$. For CBW, the mean uncertainty was $\pm45\%$ with individual values ranging from $\pm13\%$ to $\pm75\%$. The upper range and hence mean value of the uncertainties can be lowered by applying stricter event selection criteria, but at the cost of reducing the dataset. The longer term mean flux can be determined much more accurately provided there are enough observations, as its uncertainty is inversely proportional to the number of observations.

For both sites, the error in the longer term mean was about $\pm15\%$ (Table 1) suggesting that our methodology is very suitable for estimating seasonal and annual regional $^{222}$Rn surface fluxes.

For both sites the results from four independent methods (SPOT-EC, soil chamber, radionuclides-based map and process based modelling) agree well. Considering that these sites have very different soil types and conditions, this is a very promising result as it suggests that the $^{222}$Rn surface flux can be relatively well constrained by our present method. The SPOT-EC method

does not provide continuous, but rather time-averaged results. It is spatially limited by the size of the footprint, although a longer period of observations can produce a higher spatial resolution (see Fig. 7. Still, verification or comparison using a soil chamber measurement series is valuable, preferably at at least two different locations inside the footprint. The SPOT-EC results are of great value to verify or calibrate $^{222}$Rn soil flux models for a given site, or to test regional atmospheric transport models.



# 5   Conclusions

We have described a new method, which we call Single Pair of Observations Technique with Eddy Covariance (SPOT-EC), to determine regional scale surface fluxes of $^{222}$Rn from ambient measurements of $^{222}$Rn activity, $CO_2$ mole fractions and $CO_2$ eddy covariance fluxes. SPOT-EC provides mean $^{222}$Rn fluxes at hourly resolution which are integrated in space (i.e. over

the footprint) and time (i.e. the duration of a given event). Short term fluxes (from single events) can be calculated with an uncertainty of about $\pm15\%$ and longer term (e.g. seasonal / annual) mean fluxes with an uncertainty of about $\pm10\%$. SPOT-EC does not provide continuous results, however good diel coverage was obtained at both sites examined here and no significant correlation was observed between the sampling time of day and the magnitude of the flux.

We have applied our methodology to observations from two stations in the Netherlands: Cabauw and Lutjewad and compared

our results with results from two independent modelling studies, as well as soil chamber measurements. For both stations, fairly good agreement was found between these four independent methods suggesting that the $^{222}$Rn soil flux can be relatively well constrained by our method.

For LUT we estimate a mean $^{222}$Rn surface flux of $(0.29 \pm 0.02)$ atoms cm$^{-2}$ s$^{-1}$. Fluxes >0.5 atoms cm$^{-2}$ s$^{-1}$ were observed to the south and southeast. For CBW we estimate a mean $^{222}$Rn surface flux of $(0.63 \pm 0.04)$ atoms cm$^{-2}$ s$^{-1}$. Lowest fluxes (0.4

to 0.6 atoms cm$^{-2}$ s$^{-1}$) were generally observed from the northeast and the highest values (>0.6 atoms cm$^{-2}$ s$^{-1}$) were observed to the southwest, where the soil type is mainly peat or river-clay respectively.

Our methodology offers a powerful tool for calibrating process-based $^{222}$Rn soil flux models, validating regional atmospheric transport models and provides better constraints for regional inversions using the $^{222}$Rn-tracer method.

*Acknowledgements.*  The authors would like to thank B. A. M. Kers, J. C. Roeloffzen, J. K. Schut, H. Been, R. E. M. Neubert, E. Kettner, P.

Jongejan, and P. van den Bulk for various technical assistance throughout this project. Ute Karstens is greatly acknowledged for providing model-based $^{222}$Rn soil fluxes for Europe available at: http://doi.pangaea.de/10.1594/PANGAEA.854715. Data from Cabauw station are kindly provided via the Cabauw Experimental Site for Atmospheric Research (Cesar) database: http://www.cesar-database.nl/. A. Manning receives support from the UK Natural Environment Research Council (NERC) as a National Centre for Atmospheric Science (NCAS) PI.





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
