# Peer review of "Inferring 222Radon soil fluxes from ambient 222Radon activity and eddy covariance measurements of CO2"

_Atmospheric Measurement Techniques, 2016_

## Referee Comment (RC1) · Anonymous Referee #1 · 1 Jun 2016

GENERAL COMMENT:

The manuscript discusses a method of estimation of radon-222 flux from the ground that is representative for an area within several tens kilometers and results of its application. The methods is interesting and remarkable because of its reversed viewpoint. It treats radon flux as a bound variable to be estimated from a flux of other gas, while radon flux is traditionally applied as a free variable, or even a constant, to estimate a flux of other gaseous/airborne material emitted from the ground. The method seems to work.

SPECIAL COMMENTS:

[Figure]

Page 5 Line no. 62, The 4-hour accumulation with the soil chamber seems to be enough to increase water content (moisture) of surface soil and to suppress radon exhalation from the ground surface, because the studied areas have so shallow water table less than 1 m depth. Are there any observation, analysis or discussion about the effect of this phenomenon in the referred material (Manohar et al.)? If not, it should be noted that the results of radon flux measured by the chamber method would be possibly underestimated.

Table 1 The error of radon flux excluded 12 values by SPOT-EC in LUT site might be 0.02 atoms cm-2 s-1, as written in the pages 8 and 15.

Discussion about Fig. 7 Each "event" has different but not independent footprint from the others'. The footprints with longer fetches as well as shorter ones contain information about the flux from the ground just under, or closer to, the towers to different degree. Thus, is there any possibility that the estimated radon fluxes for more distant areas from the towers are, in the cases of this study, underestimated? The authors might be recommended to clarify this attention in this manuscript. It is agreed that the tendency of radon flux distributions expressed in Fig. 7 is qualitatively correct.

Page 14 Line no. 24: Typing mistakes break the sentence.

TECHNICAL CORRECTIONS:

Page 2 Line no. 21: The notation of "222Radon" and "226Radium" should be as "Radon-222" and "radium-226", respectively.

Figure 1 Some expression of the scale of the map is desired.

Page 4 Line no. 22: "Point" would be "Pair", as the authors call it in the abstract and the conclusion.

---

## Referee Comment (RC2) · I. Levin (Referee) · 18 Jul 2016

Van der Laan and co-workers present a new method to estimate soil 222Radon exhalation rates from two areas in the Netherlands. They use co-located measurements of atmospheric 222Radon activity concentration and CO2 mole fraction together with eddy-covariance flux observations of CO2 to estimate mean 222Radon exhalation rates in the catchment areas of the Cabauw and Lutjewad monitoring stations. With this top-down approach they partly overcome the lack of representativeness of localized bottom-up flux observations with soil chambers and the problem of upscaling these data to the relevant area. In this respect, their 222Radon exhalation rate estimates are more representative of the total catchment of the measurement sites than direct

chamber measurements. However, the method only works during certain meteorological situations, i.e. when significant concentration changes occur in the boundary layer, which they use for integration. These could be during nocturnal inversions, that build up mainly during stable (summer) nights, or when the air mass changes. I can, thus, imagine and this is also noted by the authors, that situations of e.g. rainfall are not well represented in the derived fluxes. However, as the authors state, during these situations fluxes can be very different (e.g. much lower due to higher soil moisture or elevated water table) than in those situations where exhalation rates can be estimated with the SPOT-EC method. This potential bias of the results is not at all mentioned in the manuscript; it may in fact also contribute to the difference to the chamber measurements at Lutjewad.

This brings me to my second major point: I am indeed wondering if the radon tracer method can be applied at these sites in the Netherlands at all and provide reliable results that are representative as annual or seasonal means. It should not be forgotten that the radon tracer method can provide valid results only under the assumption that the radon exhalation rate is more or less CONSTANT or varies only (systematically) e.g. on seasonal timescale. This is not at all the case in the study area, where the driving parameters change rapidly, as the authors state in their manuscript. Karstens et al. (2015) showed that e.g. a water table change between 0.2 and 1 meter below ground causes potential radon flux changes by a factor of three (note that 1m is the average water table depth artificially maintained in the Lutjewad and Cabauw catchment areas). If the fluxes obtained with this new method cannot be applied to other meteorological situations, the whole approach appears to be governed by circular reasoning. It is thus not clear to me why the method presented here is "more suitable for non-constant surface fluxes".

A critical discussion of these points is required in the revised version of the manuscript.

Special remarks

[Figure]

Abstract:

Line 12: Why should radon flux from peat soils be high? Line 16: How can this top-down method give "new insights in the driving mechanisms"? I would think that this would only be possible with bottom-up flux measurements, where the local parameters can be measured in addition to the exhalation rate.

Page 2:

Line 15: Give reference here that "...the radium content is relatively well know", what does it mean quantitatively? Line 17: Give reference to measurements of the large variability of the flux (orders of magnitude!) Line 24: Why is a flux chamber measurement more representative of the local radon flux than that derived from a measured soil profile? When measuring the profile, the steady state assumption/condition is still valid while measuring with a flux chamber changes the driving gradient at the soil-atmosphere interface, and there with the flux. See also comment on this point below. Line 29 and following: It should read "222Rn activity CONCENTRATION" throughout the manuscript.

Page 3:

lines 9-10: "This version ... fetch range" I do not understand this sentence. Line 12: What does this mean "...are observed at (local) background levels ..."

Page 4:

Eq. (1): Here it is assumed that the concentration C(t) is vertically constant over the mixing height h. However, this is not the case in reality and also the mixing height h is not known; h is in fact (formally) different when measurements are made at a different height (because of the concentration gradient). Therefore, I would hesitate to write the balance equation in this explicit form without further explanations.

Page 5:

Line 13: Please be precise: 222Radon is a (noble) gas, there are no 222Radon particles. If you mean radon progeny, they should be named as such. Line 15: In the Radon ICP report for the InGOS project, the uncertainty of the ANSTO measurements at an activity concentration of 1 Bq/m3 was given as 11%, please clarify, which concentration range is meant here when referring to a precision of 5%. Line 23: The accumulation chambers for radon flux measurement are kept for 4 hours (!) until the measurement of accumulated radon starts. How are the results corrected for the change in gradient or has it been tested that the increase under the chamber stayed linear over these 4 hours? There may be a systematic underestimation of the flux with these long accumulation times. Please clarify.

Page 6ff, Figure 2-8: Please change capital A, B, C to small letters in the figures or vice versa throughout the text and figure captions.

Page 7:

Line 5: Please explain what is meant with "... that our EC measurements are represented by the concentration changes..." Lines 8-9: Results were only accepted for dry periods: See my major comment on the representativeness of the derived fluxes!

Page 9:

Line 5: Do you mean "the cell south of ..."? Line 13: Should read "erroneous"

Page 11, Figure 7: Why has the marine sector been taken out? It would have been a good test of zero flux.

Page 12:

Line 17: Should better read instead of "because of rainfall", "because of soil moisture and/or water table changes during rainfall."

Page 13, line 12-15: Please explain if EC measurements were available throughout the whole time period that was selected based on the concentration measurements or

what is meant with ". . . our method does not provide semi-continuous results."

Page 14, Figure 8: I imagine that good coverage throughout the day is mainly obtained during winter and less so during summer. Please specify. Whole discussion on this page 14 needs critical assessment in view of my major remarks at the beginning of the review!

―――――――――――――――――――

---

## Author Comment (AC1) · 22 Sep 2016

First we would like to sincerely thank this reviewer for his/her comments on our manuscript. Please see our replies below:

(1)
*Page 5 Line no. 62, The 4-hour accumulation with the soil chamber seems to be enough to increase water content (moisture) of surface soil and to suppress radon exhalation from the ground surface, because the studied areas have so shallow water table less than 1 m depth. Are there any observation, analysis or discussion about the effect of this phenomenon in the referred material (Manohar et al.)? If not, it should be noted that the results of radon flux measured by the chamber method would be possibly underestimated.*

The 4 hour accumulation period seems somewhat long indeed but this was chosen because of the (expected) low radium content in the soils. Furthermore the soil around our site is made up of compacted "marine" clay which, although porous, has a relatively low permeability and hence water flow is very low. We do not expect our measurements are underestimated significantly because we measured the soil water content 0.3 m below the chamber and did not observe any sudden increases in moisture content. Of course, the measured soil water content as well as the measured Radon flux might not be representative for the much larger region as measured by SPOT-EC. But that is actually a main point of our work. We added this to the discussion section of our manuscript and included an explanation on the accumulation period in the text..

(2)
*Table 1 The error of radon flux excluded 12 values by SPOT-EC in LUT site might be 0.02 atoms cm-2 s-1, as written in the pages 8 and 15.*

Correct (typo). Modified text.

(3)
*Discussion about Fig. 7 Each "event" has different but not independent footprint from the others'. The footprints with longer fetches as well as shorter ones contain information about the flux from the ground just under, or closer to, the towers to different degree. Thus, is there any possibility that the estimated radon fluxes for more distant areas from the towers are, in the cases of this study, underestimated? The authors might be recommended to clarify this attention in this manuscript. It is agreed that the tendency of radon flux distributions expressed in Fig. 7 is qualitatively correct.*

This is a good point. This was actually the rationale behind the original SPOT method, of which the SPOT-EC method is an extended version. With this method the flux for is calculated from (a collection of) single observation points instead of a regression fit through all observations in a longer period (e.g. diurnal or weekly). Using the regression fit method, nearby emissions would be oversampled whereas with the SPOT method we actually get the 'real' mean flux per event which is representative for the area covered by the sampled air mass. For this paper we have included all events that are thought to be representative for the area around our sites (using the selection criteria described in Sect. 2.3) in the calculation of the regional mean value. In principal we could also have calculated the mean value using only events representative for the mean flux of a defined distance from the tower. In that case we would have e.g. used only those events representing 20 km and further from the site (based on event duration times wind speed) but we choose to also include the events covering shorter distances because: (1) otherwise we would have had less data to calculate the mean value from, (2) the estimation of the covered area is only a rough estimate and (3) we wanted to compare our results with chamber measurements which were performed directly next to the towers.

Note also that polar plots in general do not provide a very robust illustration of the distance of

the emissions for a similar reason: in the case of a large emission source nearby the tower, also the points on the edges of the plot will be affected, because they represent the average emissions for the complete coverage of the event (e.g. up to 60 km from the site). High values at the edge of the plot might therefore wrongfully suggest that there are large emission sources at the end of the footprint. This was also discussed by van der Laan et al. (2014).

(4)
*Page 14 Line no. 24: Typing mistakes break the sentence.*
*TECHNICAL CORRECTIONS:*
*Page 2 Line no. 21: The notation of "222Radon" and "226Radium" should be as "Radon-222" and "radium-226", respectively.*
*Figure 1 Some expression of the scale of the map is desired.*
*Page 4 Line no. 22: "Point" would be "Pair", as the authors call it in the abstract and the conclusion.*

These comments were addressed already before the online discussion.

**References**

van der Laan, S., van der Laan-Luijkx, I. T., Zimmermann, L., Conen, F., and Leuenberger, M.: Net CO2 surface emissions at Bern, Switzerland inferred from ambient observations of CO2, delta(O-2/N-2), and (222) Rn using a customized radon tracer inversion, Journal of Geophysical Research-Atmospheres, 119, 1580-1591, 10.1002/2013jd020307, 2014.

---

## Author Comment (AC2)

Dear prof. Levin, thank you very much for taking the time to comment on our manuscript. We appreciate your careful review and which allows us to improve our manuscript. Please see our replies to your comments below.

(1)
*.. the method only works during certain meteorological situations, i.e. when significant concentration changes occur in the boundary layer, which they use for integration. These could be during nocturnal inversions, that build up mainly during stable (summer) nights, or when the air mass changes. I can, thus, imagine and this is also noted by the authors, that situations of e.g. rainfall are not well represented in the derived fluxes. However, as the authors state, during these situations fluxes can be very different (e.g. much lower due to higher soil moisture or elevated water table) than in those situations where exhalation rates can be estimated with the SPOT-EC method. This potential bias of the results is not at all mentioned in the manuscript; it may in fact also contribute to the difference to the chamber measurements at Lutjewad.*

In the manuscript we explain that both the chamber and the SPOT-EC method do not provide continuous observations. The chamber provides only two 4-hourly integrated observations per day (Sect. 2.2.1) and the SPOT-EC method only works for conditions with enough turbulence to generate Eddies as well as enough atmospheric stability to provide 'measurable' concentration differences (Sect. 4). Therefore, the calculation (and hence comparison) of the mean values will be subjected to the amount of available data points as well as their temporal distribution (shown in Figure 5).

The $^{222}$Rn flux itself is not sensitive to meteorological conditions, except for soil moisture. The resulting fluxes do not depend on the degree of turbulence. This is also shown in Figure 8 which indicates there is no significant correlation between the magnitude of the flux and the time of the day. The degree of turbulence may have an impact on the uncertainty of the observations (explained in Sect. 4). For example, the relative error in the concentration measurements will be larger for relatively unstable conditions because of a lower signal to noise ratio.

Yes, rainfall will increase soil moisture content and therefore decrease the magnitude of the $^{222}$Rn surface flux. This will affect both chamber measurements as well as concentration measurements, and is not problem for our method since also the measured concentration differences will be proportionally smaller. It is true that with SPOT-EC we do not measure during rainfall but also the soil chamber measurements are not directly affected by rainfall, since the chamber is closed during the actual measurements of the flux. Both systems do measure the flux from the wetter soil after a rainfall event. The only difference is that SPOT-EC measures the flux for the wet soil *after* the rainfall whereas the chamber might also have a few samples during the rainfall when the soil is getting still wetter (assuming the soil underneath the chamber is affected by the moisture level of the surrounding soil). It is therefore unlikely that these instances are a cause for any significant bias between the two systems. The main cause for the observed differences between the methods is related to the footprints of the observations (e.g. potentially including different vegetation, ditches, soil types etc.) which is exactly what makes SPOT-EC so valuable, since it represents a much larger area than the soil chamber. Nevertheless we agree we should include any possible cause for potential bias in our discussion and have added this information to the text.

(2a)
*This brings me to my second major point: I am indeed wondering if the radon tracer method can be applied at these sites in the Netherlands at all and provide reliable results that are representative as annual or seasonal means. It should not be forgotten that the radon tracer method can provide valid results only under the assumption that*

*the radon exhalation rate is more or less CONSTANT or varies only (systematically) e.g. on seasonal timescale. This is not at all the case in the study area, where the driving parameters change rapidly, as the authors state in their manuscript.*

This is an interesting point although not directly related to our method (SPOT-EC) since with this method we can actually measure (i.e. with EC-$CO_2$) the variability in the surface flux on shorter timescales, and that is used for scaling the concentration differences. The reviewer refers to the use of the 'standard' radon tracer method (e.g. Levin, 1987;Biraud et al., 2000;van der Laan et al., 2009) and not to the method (SPOT-EC) presented in this work.

With both methods concentration changes observed at intake of a specie of interest are scaled by a factor which represents the atmospheric mixing and dilution. This factor is calculated from a known surface flux of a second specie dived by its observed concentration changes at the intake. With the 'standard' radon tracer method, observations of [222]Rn concentrations are used together with an assumed to be well-known value for the [222]Rn surface flux to calculate the atmospheric mixing component. Currently (i.e. prior to the work presented here), this [222]Rn flux is however at the most a well-educated guess based on extrapolating (in space) values from soil chambers or interpolating (in time) estimates from [222]Rn flux maps. Since the method requires the mean [222]Rn flux during the observation period (e.g. for the duration of an event) to be well-known, [222]Rn is therefore generally assumed to be constant.

With SPOT-EC this limitation does not apply since we *actually measure* the surface flux of the specie we use (i.e. $CO_2$) concurrently with its concentrations at intake, to estimate the atmospheric mixing component.

It remains difficult to speculate on the degree of uncertainty in the 'standard' radon tracer method related to temporal variability in the [222]Rn surface flux. For long term means we expect this uncertainty to be minor because of the good agreement between our yearly mean results and those from independent methods such as from flux maps and models. In any case (not just for the Netherlands) when applying the [222]Rn tracer method for shorter periods obviously one needs to take into account an additional uncertainty due to the temporal variability of [222]Rn.

We have included this in our modified discussion section to highlight the advantages of our method and to remind the reader to be cautious when using the standard [222]Rn method for relatively short time scales because of the potential variability in [222]Rn surface fluxes. One potential solution to reduce such uncertainty might be to not use the standard method for periods when the [222]Rn variability can be expected to be large, such as during rainfall.

*(2b)Karstens et al. (2015) showed that e.g. a water table change between 0.2 and 1 meter below ground causes potential radon flux changes by a factor of three (note that 1m is the average water table depth artificially maintained in the Lutjewad and Cabauw catchment areas).*

Although the [222]Rn flux is very sensitive to the amount of soil moisture, we do not expect a rapidly changing flux due to a change in water table, during the typical length of our observed events. We measured the soil water content at LUT 0.3 m below the chamber and did not observe any sudden increases in moisture content. Furthermore the water table is generally much lower then -0.2 m. For example at LUT the water table is around -1m  in winter and -1.5 m for the rest of the year (Manohar et al., 2016).

(2c)
*If the fluxes obtained with this new method cannot be applied to other meteorological situations, the whole approach appears to be governed by circular reasoning.*

This is only true if the $^{222}$Rn flux estimated with SPOT-EC is subsequently used in the 'standard' radon tracer inversion for the same specie (in this case $CO_2$) and for the same time period. However our method is not set up for that purpose, and the resulting $^{222}$Rn fluxes can be used in other applications as well as for other periods (compared to e.g. $CO_2$, the $^{222}$Rn flux is relatively homogeneously spread).

For example when using Radon to test or calibrate the atmospheric mixing component in atmospheric transport models, it is important to use the $^{222}$Rn concentration changes at intake as well as the regional surface $^{222}$Rn flux driving this concentration change. The latter is currently only roughly estimated using models or a fixed value is applied. Our method actually provides the observed $^{222}$Rn surface flux for the period of the observed $^{222}$Rn concentration change.

When using the SPOT-EC results for the 'standard' radon tracer inversion method one can either: (1) use a third specie, or (2) extrapolate the results:

(1): since the technique is equally applicable to any other measured species one can also determine the regional $^{222}$Rn flux with SPOT-EC using (e.g.) EC-$CH_4$ and then calculate the regional $CO_2$ flux from there for the same periods.

(2): because the footprint of the SPOT-EC method is constrained by the Eddy-Covariance measurements, the footprint is relatively small ($\sim$15 km distance at 60 m measurement height). The resulting $^{222}$Rn flux therefore represents this area, whereas the 'standard' Radon tracer inversion is used during purely stable atmospheric conditions representing a much larger footprint (potentially > 100 km) For such applications the $^{222}$Rn flux is therefore extrapolated in space and time and of course, depending on measurement area an uncertainty due to such extrapolation is involved. There is no apparent reason why the $^{222}$Rn flux could not be extrapolated to other periods or for larger regions. Except for soil moisture, the $^{222}$Rn flux is not very dependent on meteorological conditions (such as atmospheric stability) and in most cases the $^{222}$Rn flux is much more constant then the specie of interest. In fact, this is the main rationale behind the original $^{222}$Rn tracer inversion method. With enough observations one could even determine the $^{222}$Rn flux for different conditions such as per wind sector, soil moisture levels, etc. or, after calibration with SPOT-EC, use a land surface/process model to further constrain the extrapolation. Our method can thus easily be seen as a step forward compared to current applications which apply a (relatively) fixed value derived from e.g. $^{222}$Rn soil maps (Karstens et al., 2015;Szegvary et al., 2007) or from soil chambers.

We have updated the discussion section and included explanation that the method is not circular reasoning, and have included examples for possible applications of our method.

(2d)
*It is thus not clear to me why the method presented here is "more suitable for non-constant surface fluxes".*

This refers to the specie of interest. As explained above, the surface flux of the tracer used to constrain the atmospheric mixing is assumed to be constant for the radon tracer method as its variability is not known. If it would be known, one could account for it. When using SPOT (from which we derive SPOT-EC as presented here), a net mean surface flux is returned by the method integrated over the duration of the event and over the area covered by the air mass during this period. When using the commonly applied linear regression fit method, nearby emissions are over-counted.

We have included a more detailed discussion involving the above comments and thank the reviewer for pointing out what we need to explain better. Please see our updated discussion section.

Special remarks
*Line 12: Why should radon flux from peat soils be high?*

Typo: this should read 'river-clay'. Changed text accordingly.

*Line 16: How can this top-down method give "new insights in the driving mechanisms"? I would think that this would only be possible with bottom-up flux measurements, where the local parameters can be measured in addition to the exhalation rate.*

We agree and have removed 'and for gaining new insights in the driving mechanisms behind $^{222}$Rn surface emissions'.

Page 2:
*Line 15: Give reference here that "...the radium content is relatively well know", what does it mean quantitatively?*

What we mean is that even though the radium content is easily determined, the radon flux is not because it depends also on much more variable factors such as soil moisture etc. We have therefore rewritten this sentence as follows: 'One complicating factor is that, although the production of $^{222}$Rn is directly related to the uniformly distributed Radium content in the soil and therefore relatively well-known, its surface flux is sensitive to e.g. soil porosity and soil moisture content.' Quantitatively this of course depends on the location (note our statement is general, we do not refer specifically to LUT or CBW here).

*Line 17: Give reference to measurements of the large variability of the flux (orders of magnitude!)*

Although orders of magnitude may not be common practice, it can easily be the case since soil emissions can go to almost zero for very wet conditions (see also Manohar et al. (2013) for the relation between the magnitude of the Radon flux versus soil moisture). Modified text to: 'Therefore, the $^{222}$Rn surface flux can be very heterogeneously spread on regional scales (e.g. because of different water table heights) and vary significantly (e.g. dropping from 100% to almost zero emission) within hours due to e.g. rain fall (Manohar et al., 2013).'

*Line 24: Why is a flux chamber measurement more representative of the local radon flux than that derived from a measured soil profile? When measuring the profile, the steady state assumption/condition is still valid while measuring with a flux chamber changes the driving gradient at the soil-atmosphere interface, and there with the flux. See also comment on this point below.*
Agreed. Removed 'which is more representative for the actual surface flux'

*Line 29 and following: It should read "222Rn activity CONCENTRATION" throughout the manuscript.*
Changed accordingly.

*Page 3:*
*lines 9-10: "This version...fetch range" I do not understand this sentence.*

Rewritten as follows: 'More specifically, we modified the so-called Single Pair of Observations Technique (SPOT) described by van der Laan et al. (2014). Compared to the commonly applied technique of using a linear regression fit on all observations in a longer time period, this version of the $^{222}$Rn tracer method is more suitable for estimating non-constant surface fluxes.'

Line 12:
*What does this mean "…are observed at (local) background levels…"*

This is explained in the same sentence: '…when the atmosphere is well-mixed, ambient concentrations are observed at (local) background levels..'
Hence we refer to these well-mixed conditions as 'local background levels' (because of the strong vertical atmospheric mixing). As explained in this section, we use these conditions as a starting/reference point from which we calculate fluxes when, during stable conditions, concentrations are increasing as they accumulate in the lower part of the boundary layer.

*Page 4:*
*Eq. (1): Here it is assumed that the concentration C(t) is vertically constant over the mixing height h. However, this is not the case in reality and also the mixing height h is not known; h is in fact (formally) different when measurements are made at a different height (because of the concentration gradient). Therefore, I would hesitate to write the balance equation in this explicit form without further explanations.*

This is, in principle, a valid point for Eq. (1) but when applied to two concurrently measured species of which one has a known surface flux (either assumed or actually measured as in our case) h is cancelled out after rearranging to Eq. (2). Then, the concentrations need only to be vertically distributed equally between the surface and the intake height at which the concentrations are measured for the duration of the event from which the flux is calculated (i.e. $t_0$-$t_x$).
We have added the following explanations in the text: 'This methodology assumes equal vertical distribution for both species between surface and at intake, e.g. no sudden chemical loss or addition for one specie. Vertical mixing (e.g. due to a changing PBL height) and dilution (e.g. due to mixing with the free troposphere) is assumed to be equal for both species and hence cancelled out. In the case of entrainment, we assume our observed background concentrations at t=$t_0$ are, to a good degree, representative for the free troposphere at the site location and potential degree of mixing is equal for both species.'
A discussion on our EC observations being at a different height is included in the discussion section.

*Page 5:*
*Line 13: Please be precise: 222Radon is a (noble) gas, there are no 222Radon particles. If you mean radon progeny, they should be named as such.*

Modified text accordingly.

*Line 15: In the Radon ICP report for the InGOS project, the uncertainty of the ANSTO measurements at an activity concentration of 1 Bq/m3 was given as 11%, please clarify, which concentration range is meant here when referring to a precision of 5%.*

Here we refer to the actual precision resulting from counting statistics (typically around 3-4 % at 1 Bq m$^{-3}$) combined with the uncertainty attributed to the accuracy of the source (about 4%). For concentration ranges representative for our selected events the precision is about 2.5% at ~2.5 Bq m$^{-3}$. However, at this point in the manuscript it might be a good idea to refer to the combined uncertainty including also the coefficient of variability of valid monthly calibration coefficients and the background count variability as suggested by the reviewer. Therefore we have modified the text to:
'The total measurement uncertainty is about 11% of the measured value at both sites (at an activity concentration of 1 Bq m$^{-3}$) including measurement precision resulting from counting

statistics ($\sim$ 3-4 %), accuracy of the source ($\sim$ 4%), the coefficient of variability of valid monthly calibration coefficients (2%), and the background count variability ($\sim$10 mBq m$^{-3}$) (Popa et al., 2011;Schmithüsen et al., 2016;Van der Laan et al., 2010).'

*Line 23: The accumulation chambers for radon flux measurement are kept for 4 hours (!) until the measurement of accumulated radon starts. How are the results corrected for the change in gradient or has it been tested that the increase under the chamber stayed linear over these 4 hours? There may be a systematic underestimation of the flux with these long accumulation times. Please clarify.*

This system was extensively tested during the initial installation and the concentration increases were found to be linear over time for at least 5 hours of accumulation for wet and dry soils. The 4 hour measurement window was chosen because of the relative low soil radium content at LUT in combination with the relative high background counts of our pylon detector. The combined measurements uncertainty of $\sim$ 20% (Sect. 2.2.1) includes also error associated due to back-diffusion.
We have included this in the text.

*Page 6ff, Figure 2-8: Please change capital A, B, C to small letters in the figures or vice versa throughout the text and figure captions.*

Modified accordingly.

*Page 7:*
*Line 5: Please explain what is meant with "...that our EC measurements are represented by the concentration changes...*

This part was indeed unclear and also not necessary, and we have removed it.

*Lines 8-9: Results were only accepted for dry periods: See my major comment on the representativeness of the derived fluxes!*

See our reply above.

*Page 9:*
*Line 5: Do you mean "the cell south of ..."?*

The cell south to the prior cell. Changed accordingly.

*Line 13: Should read "erroneous"*

Removed "heterogeneous".

*Page 11, Figure 7: Why has the marine sector been taken out? It would have been a good test of zero flux.*

This is an interesting idea however the area as seen from the tower starts with a 1-2 km stretch of reclamation area with salt tolerant grasses followed by tidal flats, and the flux is therefore not zero.

*Page 12:*
*Line 17: Should better read instead of "because of rainfall", "because of soil moisture and/or water table changes during rainfall."*

Changed accordingly.

*Page 13, line 12-15: Please explain if EC measurements were available throughout the whole time period that was selected based on the concentration measurements or what is meant with "…our method does not provide semi-continuous results."*

This refers only to the $^{222}$Rn flux results (figure 5). EC measurements where available throughout the whole time as shown in figure 2 and figure 3. Modified text to:' *does not provide semi-continuous results for the estimated $^{222}$Rn fluxes*"

*Page 14, Figure 8: I imagine that good coverage throughout the day is mainly obtained during winter and less so during summer. Please specify. .*

Yes since turbulent conditions are more frequent in summers. See also figure 5. We have indicated this in the text accordingly.

*Whole discussion on this page 14 needs critical assessment in view of my major remarks at the beginning of the review*

These remarks were appreciated very much as they have helped to improve the manuscript greatly. Please see our replies above and our modified discussion section.

**References**

Biraud, S., Ciais, P., Ramonet, M., Simmonds, P., Kazan, V., Monfray, P., O'Doherty, S., Spain, T. G., and Jennings, S. G.: European greenhouse gas emissions estimated from continuous atmospheric measurements and radon 222 at Mace Head, Ireland, Journal of Geophysical Research-Atmospheres, 105, 1351-1366, 2000.

Karstens, U., Schwingshackl, C., Schmithüsen, D., and Levin, I.: A process-based 222Rn flux map for Europe and its comparison to long-term observations, Atmos. Chem. Phys., 15, 12845-12865, doi:10.5194/acp-15-12845-2015, 2015.

Levin, I.: Atmospheric $CO_2$ in continental Europe - an alternative approach to clean air $CO_2$ data, Tellus B, 39, 10.3402/tellusb.v39i1-2.15320, 1987.

Manohar, S. N., Meijer, H. A. J., and Herber, M. A.: Radon flux maps for the Netherlands and Europe using terrestrial gamma radiation derived from soil radionuclides, Atmospheric Environment, 81, 399-412, 10.1016/j.atmosenv.2013.09.005, 2013.

Manohar, S. N., Meijer, H. A. J., Neubert, R. E. M., Kettner, E., and Herber, M. A.: Radon flux measurements at atmospheric station Lutjewad – analysis of temporal trends and soil and meteorological variables influencing the emissions, in Prep., 2016.

Popa, M. E., Vermeulen, A. T., van den Bulk, W. C. M., Jongejan, P. A. C., Batenburg, A. M., Zahorowski, W., and Rockmann, T.: H-2 vertical profiles in the continental boundary layer: measurements at the Cabauw tall tower in The Netherlands, Atmospheric Chemistry and Physics, 11, 6425-6443, 10.5194/acp-11-6425-2011, 2011.

Schmithüsen, D., Chambers, S., Fischer, B., Gilge, S., Hatakka, J., Kazan, V., Neubert, R., Paatero, J., Ramonet, M., Schlosser, C., Schmid, S., Vermeulen, A., and Levin, I.: A European-wide 222Radon and 222Radon progeny comparison study, Atmos. Meas. Tech. Discuss., 2016, 1-26, 10.5194/amt-2016-111, 2016.

Szegvary, T., Leuenberger, M. C., and Conen, F.: Predicting terrestrial Rn-222 flux using gamma dose rate as a proxy, Atmospheric Chemistry and Physics, 7, 2789-2795, 2007.

van der Laan, S., Neubert, R. E. M., and Meijer, H. A. J.: Methane and nitrous oxide emissions in The Netherlands: ambient measurements support the national inventories, Atmospheric Chemistry and Physics, 9, 9369-9379, 10.5194/acp-9-9369-2009, 2009.

Van der Laan, S., Karstens, U., Neubert, R. E. M., Van der Laan-Luijkx, I. T., and Meijer, H. A. J.: Observation-based estimates of fossil fuel-derived $CO_2$ emissions in the Netherlands using Delta 14C, CO and 222Radon, Tellus Series B-Chemical and Physical Meteorology, 62, 389-402, 10.1111/j.1600-0889.2010.00493.x, 2010.